# Modeling and Validation of an Ultra-Compact Regenerative Liver Dialysis Device

**DOI:** 10.3390/bioengineering10060706

**Published:** 2023-06-11

**Authors:** Tamara Boscarino, Leone Mazzeo, Franca Abbruzzese, Mario Merone, Vincenzo Piemonte

**Affiliations:** 1Unit of Intelligent Health Technologies, Sustainable Design Management and Assessment, Faculty of Engineering, University Campus Biomedico of Rome, Via Alvaro del Portillo, 21, 00128 Rome, Italy; 2Unit of Chemical-Physics Fundamentals in Chemical Engineering, Faculty of Science and Technology for Sustainable Development and One Health, University Campus Bio-Medico of Rome, Via Alvaro del Portillo, 21, 00128 Rome, Italy; l.mazzeo@unicampus.it (L.M.); v.piemonte@unicampus.it (V.P.); 3Unit of Tissue Engineering, Faculty of Engineering, University Campus Bio-Medico of Rome, Via Alvaro del Portillo, 21, 00128 Rome, Italy; f.abbruzzese@unicampus.it; 4Unit of Computer Systems and Bioinformatics, Faculty of Engineering, University Campus Bio-Medico of Rome, Via Alvaro del Portillo, 21, 00128 Rome, Italy; m.merone@unicampus.it

**Keywords:** liver, regenerative albumin dialysis, compact artificial liver device, micro-adsorbent particle, bilirubin adsorption

## Abstract

The availability of a wearable artificial liver that facilitates extracorporeal dialysis outside of medical facilities would represent a significant advancement for patients requiring dialysis. The objective of this preliminary investigation is to explore, using validated mathematical models based on in vitro data, the feasibility of developing a novel, cost-effective, and highly compact extracorporeal liver support device that can be employed as a transitional therapy to transplantation outside of clinical settings. Such an innovation would offer substantial cost savings to the national healthcare system while significantly improving the patient’s quality of life. The experimental components consisted of replacing traditional adsorbent materials with albumin-functionalized silica microspheres due to their capacity to adsorb bilirubin, one of the toxins responsible for liver failure. Two configurations of the dialysis module were tested: one involved dispersing the adsorbent particles in dialysis fluid, while the other did not require dialysis fluid. The results demonstrate the superior performance of the first configuration compared to the second. Although the clinical applicability of these models remains distant from the current stage, further studies will focus on optimizing these models to develop a more compact and wearable device.

## 1. Introduction

The liver is a complex organ responsible for carrying out multiple functions, such as the control of metabolic processes, the synthesis of plasma proteins, and the purification of blood from endogenous and xenobiotic substances [1]. Replacing, albeit temporarily, the plurality of functions performed by the liver is one of the great challenges facing medicine [2]; therefore, transplantation is the only effective treatment for patient survival in the case of many terminal liver diseases, acute liver failure, and some metabolic or congenital diseases that involve the liver. Liver failure can occur either from an injury to an otherwise normal liver (acute liver failure, ALF) or a new injury to an already diseased liver (acute-on-chronic liver failure, ACLF). These conditions have mortality rates of about 50–80%, and death is often related to the development of multiple extrahepatic organ dysfunction and failure. This is elucidated by the observations that a dysregulated inflammatory response appears to be the central process involved in the pathogenesis of this disease, leading to multiple organ dysfunction in addition to severe liver dysfunction [3]. In Italy, where terminal liver diseases represent an important public healthcare issue and mortality from liver diseases, although slowly decreasing, is still very significant; liver transplantation has been increasing gradually, with an annual average that exceeds 1000 transplants [4]. In parallel to the increase in performed transplants, there has been an increase in the number of patients waiting for transplantation, with a mortality rate of 7.2% and an average waiting time of ~1.5 year [5]. Given the long waiting times for organ transplantation, it is of fundamental importance to patient survival to develop new and more efficient devices able to compensate for the functional deficiencies of the damaged organ while waiting for transplantation. Towards the end of the last century, some hepatic support devices that partially replaced the detoxification function of the liver and improved various biochemical parameters were developed [6]. In this sense, liver support devices are intrinsically intended for the treatment of liver disease, not on an ongoing basis but rather as a supportive therapy while waiting for a transplant or liver regeneration (where possible). Today, biosynthetic functions can be performed by a new type of bioartificial device that includes the presence of hepatocytes [7,8,9]. However, the huge costs and, above all, the considerable obstacles encountered in the development of these systems impede their diffusion in clinical practice [10]. The artificial extracorporeal liver support devices developed to date are based on hemoperfusion systems, which provide direct blood contact with adsorbent particulates with consequent hemocompatibility problems, plasma replacement, which involves the loss of important substances such as coagulation factors and thrombocytes, as well as high costs and hemodialysis, which involves toxins–blood-to-dialysis fluid passage through a membrane. In cases of hepatic insufficiency, the endogenous toxins that need to be eliminated are closely bound to plasma proteins. Since the membrane used in dialysis is impermeable to these proteins, it becomes necessary to employ albumin dialysis methods. These methods involve the use of albumin-functionalized membranes, along with a dialysis fluid that is abundant in this protein [11]. This approach enables the transfer of toxins from the blood compartment to the dialysis fluid compartment, allowing for their removal [12]. The costs of an albumin dialysis treatment are extremely high due to the high production costs of human serum albumin; therefore, over the years, various solutions have been proposed that provide regeneration mechanisms of the dialysis fluid, which minimize the quantity of albumin required for the treatment without affecting the detoxification efficiency of the system. Among the most used regenerative liver dialysis devices, the Molecular Adsorbent Recirculating System (MARS) will be taken as a reference in the evaluation of the performance of the devices proposed in the present work [13,14,15,16]. To date, hepatic dialysis treatments include therapeutic cycles that can reach up to 6 h to be carried out in specialized hospital facilities with high costs for the healthcare system [17,18,19,20]. It was suggested that if the MARS was available more widely and the method was routinely used, it would become cheaper [21]. The MARS is usually well tolerated; however, the only consistent adverse effect is thrombocytopenia. From a clinical point of view, the patient’s outcome depends very much on the etiology of the hepatic failure. Survival rates at the time of detection varied between 60% and 70% in patients with either ALF or A-on-C LF of various etiologies. In a literature study in which MARS therapy was used to treat 34 patients with ALF of various etiologies, the survival rate was 88% at 6 months and 84% at 1 year. The highest incidence of liver recovery was observed in patients with ALF caused by intoxication [3]. In order to solve these problems, intensive research activities are needed, as has been conducted for other artificial organs. For example, the availability of a wearable artificial kidney (WAK) that provides dialysis outside the hospital would be an important advancement for dialysis patients and for reducing costs [22]. There are different urea removal strategies for application in a wearable dialysis device from both a chemical and a medical perspective [23,24,25,26,27,28,29]. In the same way, the purpose of this preliminary study is to investigate, through mathematical models validated from in vitro data, the possibility of developing a new economic and ultra-compact extracorporeal liver support device that can be used outside the clinic as a bridge therapy to transplantation, offering significant savings to the national healthcare system, as well as significant benefits to the patient’s quality of life. Although various toxins are targeted for removal in the treatment of liver failure [30], this paper specifically analyzes the detoxification of bilirubin from artificial blood. Bilirubin poses a significant challenge for removal due to its high binding affinity to albumin, with reported binding constants in the range of $10^{7}$–$10^{8}$ $M^{-1}$ [31]. In order to address this, the traditional adsorbent materials were replaced with albumin-functionalized silica microspheres in the experimental setup. These microspheres were chosen for their ability to adsorb bilirubin effectively. As mentioned earlier, bilirubin serves as a crucial toxin marker for assessing the performance of artificial devices. The study primarily focused on evaluating the device’s capacity to remove bilirubin from the bloodstream, considering different configurations incorporating silica microspheres.

## 2. Materials and Methods

### 2.1. Materials

Bovine serum albumin (fraction V, defatted) and bilirubin (mixed isomers) were utilized in their original form without any additional purification steps. The 3-Aminopropyl-functionalized silica gel (40–63 µm), a white powder consisting of silica microparticles with a pore diameter of 60 Å, was chosen as the solid support to adsorb bilirubin, using albumin as a host. To aid in the functionalization (chemisorption) of the silica particles with albumin (also named regenerative beads), promoters such as MES hemisodium salt, N-(3dimethylaminopropyl)-N′-etilcarbodiimmide (EDC) and N-Hydroxysuccinimide (NHS) were utilized. Sigma Aldrich was the source of all the materials and chemicals used, which were of reagent-grade quality. In this work, the experimental activity was conducted in two different setups (described later), employing hollow fiber membrane modules. In both configurations, we used two small hollow fiber modules obtained from a renal dialysis device, the Revaclear Capillary Dialyzer 300 (Baxter International, Deerfield, IL, USA). Such fibers were characterized by a porous membrane in polyarylethylsulfone (PAES) and polyvinylpyrrolidone (PVP). The latter polymers were selected for their excellent mechanical and thermal properties. Furthermore, their biocompatibility minimizes the possibility of thrombotic phenomena or non-physiological alterations of the hemodynamic. An overview of the main characteristics of the Revaclear Capillary Dialyzer 300 is provided in Table 1.

### 2.2. Methods

The artificial blood utilized in this study consisted exclusively of albumin and bilirubin. Bilirubin, which exhibits low solubility in water, was incorporated into the solution by initially dissolving it in a 20 mM aqueous NaOH solution. Subsequently, the albumin solution was added, which had been prepared in a 0.15 M phosphate buffer (PBS) at a pH of 7.4. The concentration of bilirubin was determined by performing spectrophotometric analysis at a wavelength of 458 nm using an Infinite M200 PRO Tecan microplate spectrophotometer (Tecan Trading AG, Männedorf, Switzerland). In the subsequent sections, a detailed description of the two experimental setups will be provided.

#### 2.2.1. Experimental Setup Provided with Dialysis Fluid

The first configuration simulates a classic module of dialysis with dialysis fluid containing albumin. More precisely, this setup involves the immersion of the hollow fiber module inside a beaker containing 65 mL of dialysis fluid rich in albumin and the suspended silica microspheres. The dialysis fluid was obtained from an albumin solution prepared with a 0.1 g albumin weight dissolved in a phosphate buffer (PBS) of 0.15 M at a pH of 7.4. The rescaled hollow fiber module was inserted into the beaker by the presence of two holes drilled into it. Then, a biphasic resin was used to anchor the module to the beaker walls, avoiding leakages. Finally, the beaker was filled with the albumin-rich dialysis fluid, in which the adsorbent silica microspheres were suspended, as shown in Figure 1.

The artificial blood containing albumin and bilirubin was continuously pumped inside the fibers using a 4020 single syringe pump (Gilson Scientific, Middleton, WI, USA). The concentration of blood entering the fibers was constant and equal to $C_{TB}^{0}$. At the end of the fibers, samples of blood were collected at regular intervals of time in order to measure the concentration of bilirubin in the blood leaving the constructed module.

#### 2.2.2. Experimental Setup Provided with Functionalized Hollow Fibers

Given the considerable adsorption capacity of the silica microspheres used [32], it was decided to investigate an innovative configuration without dialysis fluid. Functionalization of the surface of a rescaled, hollow fiber module with adsorbent microspheres was performed in this way, the necessity of the dialysis fluid is excluded, and the proposed module is suitable to be used as a stand-alone regenerative albumin dialysis module. In this experimental setup, the dialysis fluid is eliminated, and the adsorbent microspheres are used to functionalize the polyarylethylsulfone and polyvinylporridone membranes, within which the artificial blood flows. Two procedures for membrane functionalization were analyzed, in particular:(a)Membrane functionalization by physical adsorption of the silica microspheres;(b)Membrane functionalization by chemical bonds with the silica microspheres.

Since the albumin–bilirubin bond is fluorescent, the fluorescence intensity was observed with a Nikon A1+R confocal microscope. Figure 2b shows the large number of chemical bonds created after the membrane functionalization.

The sample immersion in a sodium–hydroxide solution and the subsequent immersion in a HCl solution were performed in the last selected procedure. The hydrolysis of the double bonds of the carbon and the oxygen of the polymer followed, with the subsequent formation of the COOH carboxylic groups. In this way, the amino group’s functionalized silica microspheres were bound to the membrane. Recalling that the microspheres were, in turn, functionalized with albumin, the proposed module is suitable to be used as a stand-alone regenerative albumin dialysis module. To sum up, the two configurations involved are graphically represented in Figure 3. Moreover, Table 2 shows all the parameters and their respective values used in the two performed experimental setups for the hollow fiber membrane modules.

## 3. Mathematical Modeling

The experimental data collected for the two described configurations were elaborated, and two mathematical models were developed, starting from both experimental setups. The unknown parameters (the global exchange coefficient between the blood and dialysis fluid, the mass transfer coefficient between the liquid and the solid phase, and the coefficient that describes the adsorbent capacity of the microspheres) were estimated from the following mathematical models.

### 3.1. Model with Dialysis Fluid

Two main steps are involved in the bilirubin (free and bound to albumin) removal from the artificial blood:-Mass transfer across the membrane to the external compartment containing the dialysis fluid, which is assumed to be perfectly mixed;-Adsorption on silica microspheres.

The experimental setup schematically described in Figure 3a was modeled by the following set of finite difference equations. Equation (1) represents the bilirubin mass balance in the blood compartment, with the boundary conditions in Equation (2):(1)$$\frac{\partial C_{TB}\left(t,z\right)}{\partial t}=-\frac{Q_{B}}{n_{f}\pi R_{f}^{2}}\cdot \frac{\partial C_{TB}}{\partial z}-\frac{2}{R_{f}}P_{\Sigma }\cdot \left(\alpha_{B}C_{TB}\left(z,t\right)-\alpha_{D}C_{TD}\left(t\right)\right)$$
(2)$$t=0\;\;\;\;\;\;\;\;\;\;\;\;\;\;\;C_{TB}=0\;z=0\;\;\;\;\;\;\;\;\;\;\;C_{TB}=C_{TB}^{0}$$

Equation (3) is the bilirubin mass balance in the dialysate with its boundary condition:(3)$$\frac{dC_{TD}\left(t\right)}{dt}=\frac{n_{f}2\pi R_{f}P_{\Sigma }}{V_{D}}\int_{0}^{L}\left(\alpha_{B}C_{TB}-\alpha_{D}C_{TD}\right)dz-k\frac{A_{TOT}}{V_{D}}\cdot \left(\alpha_{D}C_{TD}\left(t\right)-\alpha_{P}C_{TP}\left(t\right)\right)$$
(4)$$t=0\;\;\;\;\;\;\;\;\;\;\;C_{TD}=0$$

Equation (5) shows the bilirubin mass balance on the silica’s adsorbent microspheres with the linear isotherm, Equation (6), and its boundary condition, Equation (7):(5)$$\rho_{P}\cdot \frac{dn_{T}\left(t\right)}{dt}=k\frac{A_{TOT}}{V_{D}}\cdot \frac{\varepsilon }{1-\varepsilon }\cdot \left(\alpha_{D}C_{TD}\left(t\right)-\alpha_{P}C_{TP}\left(t\right)\right)$$
(6)$$n_{T}\left(t\right)=m^{\prime }\cdot C_{TP}\left(t\right)$$
(7)$$t=0\;\;\;\;\;\;\;\;\;\;\;n_{T}=0$$

Being ε, the void fraction volume is defined as:(8)$$\varepsilon =\frac{V_{D}}{\frac{M_{P}}{\rho_{P}}+V_{D}}$$
and $m^{\prime }$ is the equilibrium ratio between m, the equilibrium constant of the isotherm [32], and the dialysate albumin concentration ($C_{ALB}^{D}$); $A_{TOT}$ is the total microsphere surface; $P_{\Sigma }$ is the global mass transfer coefficient that takes into account both the liquid film resistances in the inner and the outer side of the membrane and resistance due to mass transport across the membrane itself. Mass balances refer to the total bilirubin concentration given by the sum of free and bound-to-albumin bilirubin, considering that only the free toxin can pass through the fiber membrane and can be adsorbed on silica microspheres. Microscopic unidimensional material balances (along the fiber axis) were considered for the blood compartment, neglecting the axial diffusive contribution. For the dialysate, a macroscopic mass balance was carried out. For this reason, in Equation (3), the integral average of free toxin concentrations in the blood and in the dialysis fluid was used. The adsorption of bilirubin (Equation (5) on silica microspheres was modeled using a linearized form of the Langmuir isotherm, coherently with the work of Annesini et al. [19]. The bilirubin mass transfer from the dialysate to the solid adsorbent was modeled using the Linear Driving Force (LDF) approximation based on the assumption that the concentration of toxins within the solid adsorbent was constant. This hypothesis (made possible by the reduced dimensions of the silica microspheres, ~25 µm) allowed us to neglect the toxin concentration profile within the microsphere and to linearize the driving force within the solid. The toxin considered for this study, bilirubin, binds to plasma albumin with a bond that has a binding constant, $k_{B}$. Only free bilirubin is able to cross the membrane, and $\alpha_{i}C_{Ti}$ with i=B,D,P represents the concentration of free bilirubin in the blood, dialysis fluid, and microsphere compartments. In particular, $\alpha_{i}$ represents the free toxin rate over the total amount:(9)$$\alpha_{i}=\frac{1}{2}\left[-\frac{C_{ALB}^{i}}{C_{T_{i}}}-\frac{1}{k_{B}C_{T_{i}}}+\sqrt{{\left(1-\frac{C_{ALB}^{i}}{C_{T_{i}}}-\frac{1}{k_{B}C_{T_{i}}}\right)}^{2}+\frac{4}{k_{B}C_{T_{i}}}}\right]$$
(10)i=B,D,P;

Assuming that $C_{ALB}^{i}\gg C_{T_{i}}$, it is possible to simplify the previous relationship into the following one:(11)$$\alpha_{i}=\frac{1}{1+k_{B}C_{ALB}^{i}}$$

In both cases:(12)$$\alpha =\frac{C_{T}}{C_{tox}}$$
where $C_{T}$ is the free toxin concentration, and $C_{tox}$ is the total toxin concentration, the sum of free and albumin-bound toxins:(13)$$C_{tox}=C_{T}+C_{TA}$$

$C_{TA}$ being the albumin-bound toxin concentration.

### 3.2. Model without Dialysis Fluid

In the second configuration, we considered a microscopic mass balance for the solid phase since the spheres are chemically bound to the outer surface of the fibers. In this case, the global material transfer coefficient includes resistance to transport relative to the liquid film on the artificial blood side and resistance relative to the membrane and transport on the microspheres. Figure 4 reports a sketch shown of the functionalized hollow fiber.

Furthermore, in this case, two main steps are involved in the bilirubin (free and bound to albumin) removal from the artificial blood:-Mass transfer across the membrane;-Adsorbed-on-silica microspheres linked to the membrane surface.

The experimental setup described has been modeled by the following set of finite difference equations. Equation (14) represents the bilirubin mass balance in the blood compartment with the boundary conditions in Equation (15):(14)$$\frac{\partial C_{TB}\left(t,z\right)}{\partial t}=-\frac{Q_{B}}{n_{f}\pi R_{f}^{2}}\cdot \frac{\partial C_{TB}}{\partial z}-\frac{2}{R_{f}}P_{\Sigma }^{\prime }\cdot \left(\alpha_{B}C_{TB}\left(z,t\right)-\alpha_{P}C_{TP}\left(t\right)\right)$$
(15)$$t=0\;\;\;\;\;C_{TB}=C_{TB}^{0}\;z=0\;\;\;\;\;C_{TB}=C_{TB}^{0}$$

As shown in Equation (15), the initial condition is different from the previous Equation (2). In this case, at the beginning of the test, the bilirubin concentration in the blood is at a maximum since the fibers are initially full of blood. This full-fiber modeling was carried out to make the analysis more accurate in order to obtain a decreasing monotonic trend caused by the saturation behavior of the solid particles. In Equation (16), the bilirubin mass balance on the silica’s adsorbent microspheres is provided together with the linear isotherm, Equation (17), and its boundary condition, Equation (18):(16)$$\rho_{P}\cdot \frac{\partial n_{T}}{\partial t}=\frac{2R_{f}}{{\left(R_{f}+\delta \right)}^{2}-R_{f}^{2}}P_{\Sigma }^{\prime }\cdot \left(\alpha_{B}C_{TB}-\alpha_{P}C_{TP}\right)$$
(17)$$n_{T}=m^{\prime }\cdot C_{TP}$$
(18)$$t=0\;\;\;\;\;n_{T}=0$$

δ being the adsorbent solid layer thickness, and $P_{\Sigma }^{\prime }$ is the overall exchange coefficient that takes into account the solid internal resistance. The same assumptions of the previous mathematical model were adopted regarding the exclusivity of free bilirubin adsorption on silica microspheres: neglecting the axial contribution of the diffusive phenomenon on one-dimensional microscopic matter balances; the use of the LDF (Linear Driving Force) model for the transfer of toxins within the solid adsorbent.

## 4. Results and Discussion

The increase in the particle adsorbent weight in the dialysis fluid was tested in the first configuration. The increased removal of bilirubin from the artificial blood can be observed due to the increased number of adsorbent particles in the dialysis fluid during the time of the experimental simulation (720 min), as shown in Figure 5. This occurs because the presence of the adsorbent beads regenerates the dialysis fluid. By increasing the useful surface area for the adsorption of free bilirubin, the concentration of free toxins in the dialysis fluid decreases, thus increasing the overall driving force, that is, the gradient of the free toxin concentration between the artificial blood and the dialysis fluid.

In order to obtain the simulated curve of the first configuration, the set of differential Equations (1)–(7), was integrated numerically with the gProms package (Process System Enterprises, London, UK), using a first-order backward finite difference method. The same software was used to perform data fitting, using k and $P_{\Sigma }$ as the adjustable parameters and following a least squares criterion. The results of the fitting together with the value of the fixed parameter ($m^{\prime })$ are given in Table 3.

Figure 6 shows the simulated curve obtained from gPROMS compared with the experimental data points of the outlet toxin concentration in the artificial blood versus time.

The model simulation follows the trend correctly with respect to the experimental data, especially from 100 min to 500 min. The trend shows a zero-toxin concentration in the blood initially because the fibers are empty. While the fiber fills up, the concentration rises sharply due to the maximum driving force between the artificial blood and the dialysate fluid. Then, the removed toxin concentration in the blood decreases, which is caused by an active site saturation between the bilirubin and the solid adsorbent in the dialysate.

Eliminating the need for dialysis fluid in such a blood purification device would make it less heavy and bulky, as well as easier to transport. This was investigated in the second configuration. In this new configuration, the set of differential Equations (14)–(18), was integrated numerically with the gProms package. The same software was used to perform data fitting, using $m^{\prime }$ and $P_{\Sigma }^{\prime }$ as the adjustable parameters (Table 4) and following a least squares criterion.

Comparing the values of the parameters estimated from the first and second configurations, it is possible to note the reduction of both the $m^{\prime }$ and $P_{\Sigma }^{\prime }$. The decrease in the former can be attributed to the modification of the adsorption capacity of the silica particles. As a matter of fact, the instauration of the chemical bonds between the membrane and the adsorbent particles reduces the number of active sites available for bilirubin adsorption. The lower value of the $P_{\Sigma }^{\prime }$ is instead obviously linked to an increase in the transport resistance since the $P_{\Sigma }^{\prime }$ also takes into account the solid internal resistance. Indeed, knowing that the transport resistance in a fluid is lower than the transport resistance in a solid, it is correct that the $P_{\Sigma }^{\prime }$ has a low-value respect to the $P_{\Sigma }$.

Figure 7 demonstrates that the model simulation accurately predicts the experimental data. As stated in the second mathematical model, the experiments and model were conducted with the assumption that the hollow fibers would be initially filled with artificial blood. This explains the curve’s trend, starting when the output and input bilirubin concentrations in the artificial blood are equal. A comparison was made between the curves of the configurations with dialysis fluid (Figure 6) and without dialysis fluid (Figure 7). The behavior of the initial concentration in the second configuration shows a slower decrease compared to the first configuration. This indicates the main disadvantages of the second device, which include:-Extended time for toxin removal from the blood;-Lower efficiency.

In the second configuration, the toxin removal from the blood reaches only 50% within 0 to 100 min, whereas in the first configuration, it averages around 78%. Generally, the first configuration exhibits higher efficiency (averaging 70%) compared to the second configuration (averaging 55%). Based on these observations, it can be concluded that the performance of the first modeled device is superior to that of the second configuration.

**Figure 7 bioengineering-10-00706-f007:**
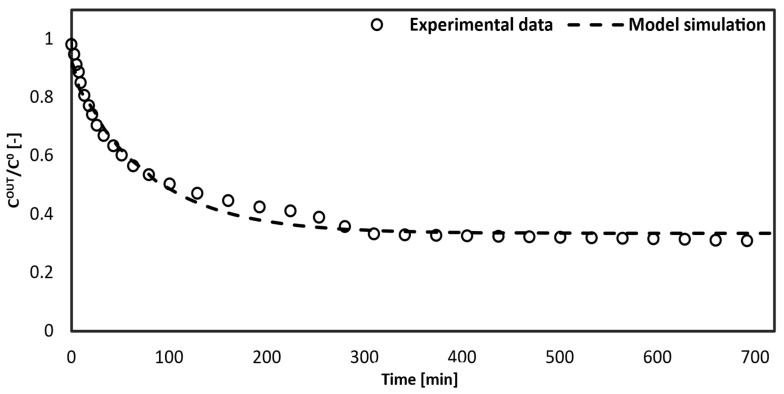
Comparison between toxin concentration in blood at the exit of the capillary and the model simulation.

## 5. Conclusions

Two configurations of small hollow fiber modules were investigated in this study. The first configuration involved immersing the hollow fibers in a dialysis fluid containing dispersed albumin-functionalized silica adsorbent particles. In the second configuration, the membrane of the fibers was chemically bound to the silica microspheres for functionalization. Mathematical models were developed for each experimental setup to describe the relevant phenomena occurring in the different configurations. The experimental data were utilized to determine the parameters for the model fitting, enabling a comparison between the simulated models and the actual experimental data. The results indicated that the simulated model accurately represented the trends observed in the experimental data for both configurations. Although the first configuration demonstrated better performance than the second one, further studies will focus on the latter configuration. The goal is to develop a compact and wearable device that can be utilized by patients, with liver disease, outside of hospital and home environments. The models shown in this study are validated by the experimental data. The mathematical model developed in this work is based on a preliminary study carried out by using bilirubin as the key toxin of the performance device. Further studies will be devoted to assessing device clearance in relation to several other toxins involved in liver failure, such as tryptophan, mercaptans, and bile acids. It is worth noting that the proposed device was tested using artificial solutions far from having the same characteristics as human blood. Therefore, the definitive application of this experimental setup in patients will require much additional research also aimed at avoiding certain possible side effects (hemolysis, cell destruction, and thrombocytopenia).

## Figures and Tables

**Figure 1 bioengineering-10-00706-f001:**
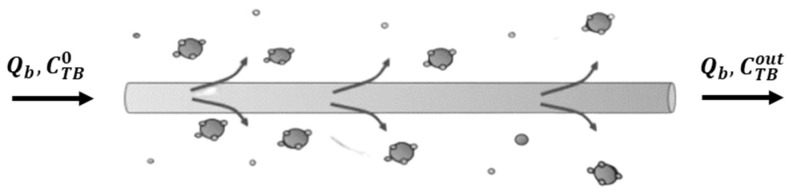
Details of the experimental setup provided with dialysis fluid highlighting the diffusion phenomena occurring between the three compartments of blood, dialysis fluid, and silica microspheres.

**Figure 2 bioengineering-10-00706-f002:**
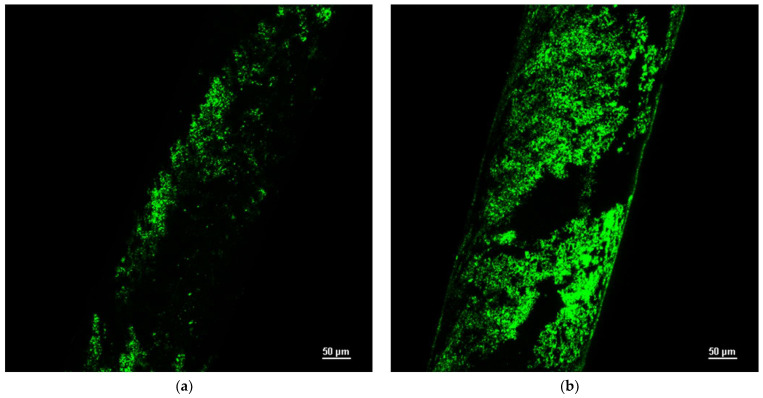
Fluorescence intensity analysis of functionalized membrane by physical adsorption (**a**) and chemical bonds (**b**).

**Figure 3 bioengineering-10-00706-f003:**
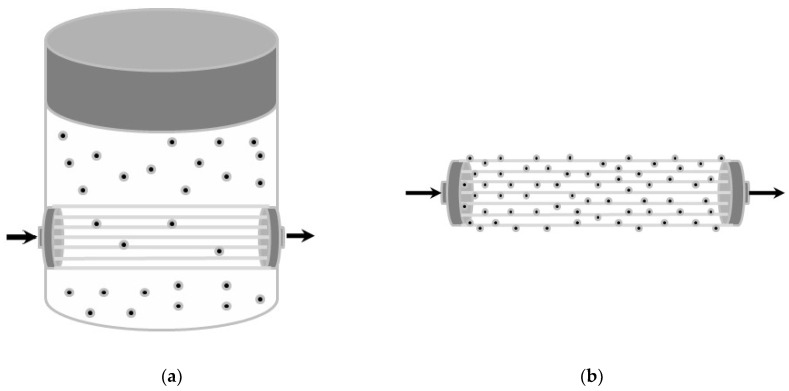
Experimental setups: (**a**) module with regenerative beads dispersed in dialysis fluid; (**b**) module with regenerative beads linked to fibers.

**Figure 4 bioengineering-10-00706-f004:**
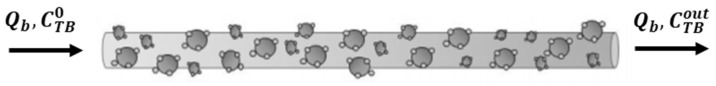
Detail of the fiber experimental setup not provided with dialysis fluid.

**Figure 5 bioengineering-10-00706-f005:**
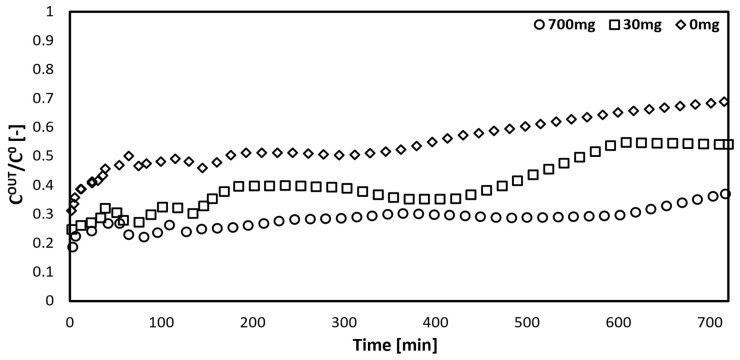
Experimental data comparison at different adsorbent masses in dialysis fluid.

**Figure 6 bioengineering-10-00706-f006:**
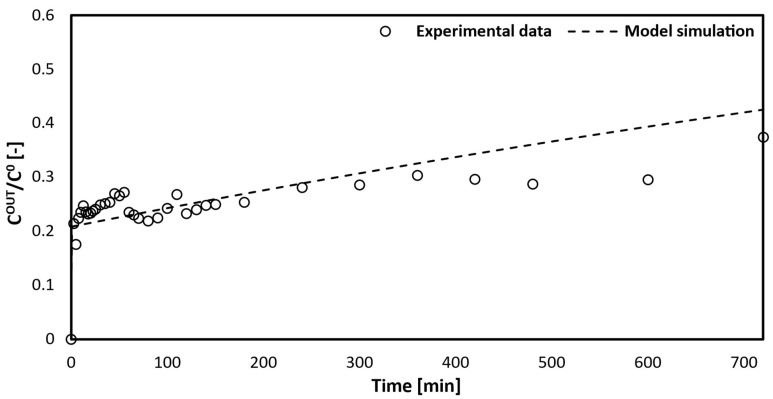
Comparison between the model curve and the experimental data collected using 700 mg of silica microspheres dispersed in the dialysis fluid.

**Table 1 bioengineering-10-00706-t001:** Main characteristics of the Gambro Revaclear Capillary Dialyzer 300.

Technical Details	Value
Surface area	1.4 $m^{2}$
Length	25 cm
Inner diameter hollow fiber	180 mm
Wall thickness membrane	35 mm
Number of fibers	15,000
Blood flow rate	200–500 mL/min
Dialysate flow	300–800 mL/min
Sieving coefficient, Albumin	<0.01

**Table 2 bioengineering-10-00706-t002:** Values used in the performed experiments with dialysis fluid (**a**) and without dialysis fluid (**b**).

Model Parameter	Description	Value (a)	Value (b)
*L*	Module length	3.5–5 cm	3.5–5 cm
$n_{f}$	Fiber number	≃45	≃45
$C_{TB}^{0}$	Total concentration of blood toxin	14.3 μmol/L	14.3 μmol/L
$C_{ALB}^{B}$	Concentration of blood albumin	1.42 μmol/L	1.42 μmol/L
$C_{ALB}^{D}$	Concentration of dialysate albumin	5.75 μmol/L	-
$M_{P}$	Adsorbent particle mass	0, 30, 700 mg	≃100 mg
$V_{d}$	Dialysate volume	65 mL	65 mL
$Q_{b}$	Blood flow rate	0.4 mL/min	0.4 mL/min
$\rho_{P}$	Particle solid density	0.2 g/mL	0.2 g/mL
t	Time	12 h	12 h

**Table 3 bioengineering-10-00706-t003:** Parameters of the first configuration.

Module Parameter	Description	Value	Reference
k	Solid exchange coefficient	2.04 cm/min	
$m^{\prime }$	Equilibrium ratio	0.33 L/g	[32]
$P_{\Sigma }$	Global exchange coefficient	57.67 cm/min	

**Table 4 bioengineering-10-00706-t004:** Parameters estimated in the second configuration.

Module Parameter	Description	Value
$m^{\prime }$	Equilibrium ratio	0.13 L/g
$P_{\Sigma }^{\prime }$	Global exchange coefficient	0.83 cm/min

## Data Availability

Not applicable.

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
