# Peer review of "Modeling and Validation of an Ultra-Compact Regenerative Liver Dialysis Device"

_bioengineering, 2023, doi:10.3390/bioengineering10060706_

Round 1

Reviewer 1 Report

This interesting and original device study focus on the adsorption of silica on one of the most used worldwide hemodialysis membrane (Revaclear) with the aim to build a wearable or a simplified device as compared to the MARS System for patients with very advanced hepatic disease awaiting liver transplantation. Two minor comments :

-first the hemodialysis dialyzer Revaclear was previously commercialized by the firm Gambro which was bought 3 years ago by Baxter ; so now Revacler dialyzer is commercialized by Baxter;

-second :authors have made their first evaluation by studying clearance of  bilirubine in an artificial blood ; could the authors describe in  a complementary discussion section the next step of their work with the analyses of epuration of other hepatic toxins namely ammoniac, biliary acids, Tryptophan and mercaptans .

A lecture of this text by a native English translator would be of interest

Author Response

Dear reviewer thank you for these comments.

-First: Sorry for the mistake, as you can find in the 'materials and methods' section, I have updated the manufacturer of the product.

-Second: As you suggested, I have included in the 'conclusion' section the next steps of this preliminary study.

Reviewer 2 Report

This is an interesting mathematical take on solving a very challenging clinical issue. However simply removing bilirubin, would not keep a patient alive with acute liver failure. Hence the significance of your work is questionable. Much more discussion regarding the survival and outcomes of current machines/devices such as the MARS, would put your work into perspective. The synthetic liver function is not being replaced, hence overwhelming coagulopathy, hypoglycaemia, hypothermia and no mention of the acute renal failure associated with acute liver failure that also occurs. Also I found it very difficult to comprehend or see your results from the two different liver dialysis models, as appears embedded within your mathematical modelling. Hence as a reader I remain sceptical as to any real benefit of either proposed device, regarding any potential clinical survival benefit.

There were some minor English errors, but otherwise fine. Very hard to read, unless by a mathematician.

Author Response

Dear reviewer thank you for these comments.

-First: As you suggested, I have included in the 'introduction' section the survival data pertaining to patients experiencing various causes of hepatic failure and some additional information on MARS.

- Second: In order to make the results more understandable, especially graphically, I have added some lines (354-358) to clarify the results obtained from the second model. Doing so, the differences between the two models were also highlighted.

- Third: As stated in the conclusion section, this paper constitutes just a first step toward the realization of a possible wearable device addressing hepatic failure. We recognize that several improvements are required to reach a complete replacement of the synthetic liver function. The ones that you outlined will be surely considered to improve our device in further studies, with the aim of clinical validation.
